# Chlorpyrifos- and Dichlorvos-Induced Oxidative and Neurogenic Damage Elicits Neuro-Cognitive Deficits and Increases Anxiety-Like Behavior in Wild-Type Rats

**DOI:** 10.3390/toxics6040071

**Published:** 2018-12-01

**Authors:** Aminu Imam, Nafeesah Abdulkareem Sulaiman, Aboyeji Lukuman Oyewole, Samson Chengetanai, Victoria Williams, Musa Iyiola Ajibola, Royhaan Olamide Folarin, Asma’u Shehu Muhammad, Sheu-Tijani Toyin Shittu, Moyosore Salihu Ajao

**Affiliations:** 1Neuroscience Unit, Department of Anatomy, College of Health Sciences, University of Ilorin, P.M.B 1515, Ilorin 240003, Nigeria; abdulkareem99mona@gmail.com (N.A.S.); moyoajao@yahoo.com (M.S.A.); 2Comparative Neurobiology Unit, School of Anatomical sciences, Faculty of Health Sciences, University of the Witwatersrand, 7 York Road, Parktown 2193, Johannesburg, South Africa; schengetanai@gmail.com (S.C.); victoriamarywilliams@gmail.com (V.W.); 3Neurophysiology Unit, Department of Physiology, College of Health Sciences, University of Ilorin, P.M.B 1515, Ilorin 240003, Nigeria; oyewole.al@unilorin.edu.ng; 4Department of Anatomy and Physiology, Faculty of Medicine, National University of Science and Technology, Bulawayo 0000, Zimbabwe; 5Institute of Neuroscience, National Yang-Ming University, Shih-Pai, Taipei 11221, Taiwan; musaiyiola@gmail.com; 6Department of Anatomy, Olabisi Onabanjo University, Ago-iwoye, Ogun State, Nigeria; royhaan.folarin@oouagoiwoye.edu.ng; 7Department of Human Anatomy, Faculty of Basic Medical Sciences, Federal University of Dutse, PMB 7156, Dutse, Jigawa State, Nigeria; Asmau.shehu@fud.edu.ng; 8Endocrinology and metabolism Research Unit, Department of Physiology, College of Medicine, University of Ibadan, Ibadan, Oyo State, Nigeria; toyinsts@yahoo.com

**Keywords:** oxidative damage, organophosphates, neurotoxicity, spatial working memory, anxiety-related behaviors

## Abstract

The execution of agricultural activities on an industrial scale has led to indiscriminate deposition of toxic xenobiotics, including organophosphates, in the biome. This has led to intoxication characterized by deleterious oxidative and neuronal changes. This study investigated the consequences of oxidative and neurogenic disruptions that follow exposure to a combination of two organophosphates, chlorpyrifos (CPF) and dichlorvos (DDVP), on neuro-cognitive performance and anxiety-like behaviors in rats. Thirty-two adult male Wistar rats (150–170 g) were randomly divided into four groups, orally exposed to normal saline (NS), DDVP (8.8 mg/kg), CPF (14.9 mg/kg), and DDVP + CPF for 14 consecutive days. On day 10 of exposure, anxiety-like behavior and amygdala-dependent fear learning were assessed using open field and elevated plus maze paradigms, respectively, while spatial working memory was assessed on day 14 in the Morris water maze paradigm, following three training trials on days 11, 12, and 13. On day 15, the rats were euthanized, and their brains excised, with the hippocampus and amygdala removed. Five of these samples were homogenized and centrifuged to analyze nitric oxide (NO) metabolites, total reactive oxygen species (ROS), and acetylcholinesterase (AChE) activity, and the other three were processed for histology (cresyl violet stain) and proliferative markers (Ki67 immunohistochemistry). Marked (*p* ≤0.05) loss in body weight, AChE depletion, and overproduction of both NO and ROS were observed after repeated exposure to individual and combined doses of CPF and DDVP. Insults from DDVP exposure appeared more severe owing to the observed greater losses in the body weights of exposed rats. There was also a significant (*p* ≤0.05) effect on the cognitive behaviors recorded from the exposed rats, and these deficits were related to the oxidative damage and neurogenic cell loss in the hippocampus and the amygdala of the exposed rats. Taken together, these results provided an insight that oxidative and neurogenic damage are central to the severity of neuro-cognitive dysfunction and increased anxiety-like behaviors that follow organophosphate poisoning.

## 1. Introduction

Indiscriminate deposition of xenobiotics into the environment has been associated with the increase in accidental poisoning and non-specific multi-organ toxicity. Oxygen stress resulting from the imbalance between the body’s antioxidant systems and the rate of free radical generation has been implicated in the pathophysiology of the subsequent toxicity from exposure to many xenobiotics as well as the development of many diseases [1,2,3,4]. Organophosphate pesticides (OPs) are a major example of xenobiotics that are indiscriminately released into the environment to control pests and insects in households and agriculture; however, their use has been linked to unavoidable concomitant diseases that result from accidental poisoning due to deposition in food substances and water, and through occupational inhalation by workers [3]. Although the primary mechanisms of OP poisoning is through irreversible binding and inhibition of acetylcholinesterase (AChE), leading to cholinergic dys-homeostasis [5], most of the destructive activities of these substances have been largely linked to oxidative damage, a widely implicated factor that complicates OP-induced toxicity [6,7,8,9,10,11,12].

In addition, OPs have been implicated in the induction of deleterious oxidative changes in various organs in the body. Their activities with respect to antioxidant free-radical balance are of vital importance, since free radicals are important mediators in the pathophysiology of most neurodegenerative diseases [13]. The neurologic effects of OP toxicity is manifested as chronic organophosphate-induced neuropsychiatric disorder (COPIND), which is characterized by cognitive deficits, depression, anxiety, and some personality problems [14,15]. All of these are associated with increased generation of reactive oxygen and nitrogen species (ROS and RNS, respectively), and/or nitric oxides in the brain, as well as anticholinesterase activity [8,9,16].

In addition, increased oxidative damage has been implicated in adversely affecting psychological and cognitive related functions through disruption of normal neurogenesis in the hippocampus and other potential hotspots within the brain [8,9,17,18,19]. The hippocampus and amygdala are two brain regions that are known to be directly involved in spatial memory and emotional reactions, such as fear. Acute and chronic exposures to either chlorpyrifos (CPF) or dichlorvos (DDVP) have resulted in a wide range of toxicities, including cardiotoxicity, neurotoxicity, hepatotoxicity, renal toxicity, hematological toxicity, and immune system toxicity, among others [8,9,20,21,22,23]. Besides cholinesterase inhibition, these substances caused marked disruptions in normal oxidative functions [8,9,20,21,24]. The use of various OPs means individuals may be exposed to more than one type of OP. It remains unknown whether co-administration of these two OPs doubles the deleterious effects observed, and which of these is a more potent poison. Thus, in this study, we investigated the neuro-cognitive consequences of uptake of two commonly used OPs, chlorpyrifos (CPF) and dichlorvos (DDVP), in rats, with possible effects on oxidative stress and proliferative functions in the hippocampus and the amygdala. DDVP and CPF have been shown to be amongst the most widely used OPs in the world [25,26].

## 2. Materials and Methods

### 2.1. Chemicals and Drugs

DDVP (PubChem Substance ID 329756736) and CPF (PubChem Substance ID 329756699) PESTANAL^®^ (analytical standard) were obtained from Sigma (Sigma-Aldrich, St. Louis, MO, USA), while the normal saline solution was prepared in our laboratory.

### 2.2. Animals and Experimental Design

A total of 32 male adult Wistar rats whose weights ranged from 150 g to 170 g were used in this study. The rats, which were obtained from the University of Ilorin biological garden, were fed with a standard rat laboratory diet and water and were kept in well-ventilated humane cages in the Faculty of Basic Medical Sciences, College of Health Sciences, University of Ilorin, Ilorin. The rats were primed to a controlled 12 h day/night cycle 7 days prior to the commencement of the experimental period. Standard guidelines were strictly adhered to in the care and use of these animals. 

### 2.3. Treatment Schedule

The rats were randomly divided into four groups (*n* = 8) as follows:Group 1 (control)—given normal saline (1 mL/kg orally) daily for 14 daysGroup 2—given DDVP (8.8 mg/kg orally) daily for 14 days [8,20,21]Group 3—given CPF (14.9 mg/kg orally) daily for 14 days [9]Group 4—given DDVP (8.8 mg/kg orally) plus CPF (14 mg/kg orally) daily for 14 days

The dosages given were with reference to previously determined LD50 values for these drugs. Previous studies have demonstrated that one-tenth of the LD50 values in rodents provide a fairly reliable estimate for projection to the human condition [27]. All experimental procedures were carried out between 07:00 and 08:30 h, and the relevant treatments were given every day over a two-week period.

### 2.4. Ethical Approval

This research work was approved by the University of Ilorin Ethical Review Committee (UERC) (UERC/ASN/2017/856), July 27, 2017, following the recommendation of the College of Health Sciences Ethical Review Committee, in compliance with the Institutional Animal Care and Use Committee (IACUC).

### 2.5. Evaluationof Brain and Body Weight

The body weights of all the rats were recorded after acclimatization on the first day of the exposures as initial weight, and on the last day of exposure as the final weight. The differences between the two weights were thus calculated and recorded as the weight changes. The brain weights of all rats were recorded after the sacrifice, and a ratio of the brain to final body weight was calculated and recorded.

### 2.6. Behavioral Evaluations

The rats were subjected to behavioral evaluations on the 14th day of the treatment to assess, short term memory, long term memory and reference memory in the Morris water maze paradigm.

#### 2.6.1. Morris Water Maze Procedure

The Morris water maze (MWM) apparatus is the most commonly used model to test spatial learning and memory. To evaluate spatial memory, rats were tested in a circle shaped black pool filled with 23–24 °C water (pool dimensions: 60 cm deep × 136 cm diameter). The pool was divided to four quadrants with boundaries labelled north (N), east (E), south (S), and west (W), and a circular platform (10 cm diameter, 28 cm high) was submerged about 2 cm below water surface in the central area of the southwest quadrant of the pool. Animals were allowed to swim until they found, mounted and remained on the platform for 15 s. If they were not able to find the platform after 60 s of swimming, they were guided to the platform by examiner and were allowed to stay on it for 15 s. The rats were then removed from the pool, dried and placed in their holding bin for 5 min. Trials were recorded by a video system. Animals received a training session consisting of three trials per session (once from each starting point) for 3 days (days 11, 12, and 13), with each trial having a maximum duration of 60 s and a trial interval of approximately 30 s. Twenty-four hours after the acquisition phase, the time taken to locate the hidden platform (escape latency) was recorded as long-term memory (LTM), and the average of the escape latency of the two subsequent trials was recorded as the short-term memory (STM). A probe test was conducted by removing the platform and allowing the rats to swim freely in the pool for 60 s; the time spent in the target quadrant which had previously contained the hidden platform was recorded as the reference memory (14th day). The time spent in the target quadrant indicated the degree of relative memory consolidation which had taken place after learning [28,29].

#### 2.6.2. Anxiety-Like Behaviors and Fear Learning

The rats were subjected to behavioral evaluations on the 13th day of the exposures to evaluate anxiety-related behaviors and fear-related learning in the open field test (OFT) and the elevated plus maze (EPM) paradigms.

##### OFT Procedure

The animals were exposed to a trial in the OFT to evaluate anxiety-related behaviors in rats following DDVP and/or CPF exposures. The rats were individually placed in the center of the apparatus, and the time spent in the center as well as the immobility period were recorded in a 5-min session. All animals were monitored in a balanced design during the procedures. For analysis, the trial was performed in a well-illuminated wooden box divided into 4 × 4 squares. It has been reported that avoidance of central squares with preference towards the perimeter areas provides an indication of elevated anxiety levels in the rats [30,31,32].

##### EPM Procedure

To evaluate amygdala-dependent or fear-related learning, the rats were exposed to two trials in the EPM paradigm. The consisted of two open arms, surrounded by a short edge to prevent falls, and two enclosed arms erected in such a way that the two open arms were opposite each other. The maze was raised about 35 cm above the ground with a stable stand and the arms of the maze were connected by a central platform. At each of the two trials, each rat was gently placed on an open arm, positioned to face away from the central platform and the closed arms. The time it took the rats to retreat and move to the closed arms was recorded as the transfer latency. While the first trial was for acquisition, the second was used as a measure of fear learning. The principle of this experiment is primarily based on the aversion of rats to heights and open spaces [9,33].

### 2.7. Biochemical Evaluation

At the end of the treatment period, the animals were euthanized with an overdose of intraperitoneal ketamine (10 mg/kg ip) and the brains were quickly dissected out and weighed. Blocks of hippocampal and amygdala tissues (from Bregma –2.5 mm to –4.5 mm) were removed from the brains of five rats from each group, dipped in 30% sucrose solution, homogenized, and portions centrifuged at 2500 revolutions per minute for 10 min. The supernatant was collected into tubes containing the reagents for the NO and ROS analysis.

ROS were measured by monitoring the increasing fluorescence of DCFH-DA following a previously described procedure using flow cytometry (Partec, Deutschland) equipped with a 488-nm argon ion laser and supplied with the Flomax software and the signals were obtained using a 530-nm band pass filter (FL-1 channel). Each determination was based on the mean fluorescence intensity of 10,000 counts [34]. The remaining tissue homogenate was added to the Griess reagents, with sulfanilamide and naphthyl ethylene diamine solutions to measure nitrate/nitrite production (NO metabolites). Absorbance was measured with the aid of a microplate reader and the levels of NO metabolites were calculated from standard curve [35]. The remaining portions of the homogenized hippocampal tissues were placed in phosphate buffer with 1% Triton-X 100 and centrifuged at 5000 rpm for 10 min. The following reagents were used: 35 μL of 5 mM dithio-bisnitrobenzoic acid, also known as Ellman’s reagent (DTNB), 10 μL of 75 mM acetylthiocholine (ATCh), and 50 mM phosphate buffer (pH 8.0). Protein concentration in brain homogenates was quantified using a Bradford assay and AChE activity was calculated in micromoles of ATCh hydrolyzed per hour per milligram of protein and was expressed as percentage of control activity and measured values in micromole per hour per milligram of protein.

### 2.8. Tissue Processing and Histopathology

After euthanasia and extraction the brains of three rats from each groups, the brains were fixed in 10% formalin for 24 h, hippocampal and amygdala blocks (from Bregma –2.5 mm to –4.5 mm) were removed, dehydrated through ascending grades of alcohol, cleared in xylene, and embedded in paraffin blocks. Every second and third hippocampal and amygdala tissues sections (5 μm in thickness) were stained with Nissl stain and/or immunostained to reveal Ki67 proliferative nuclei protein, analyzed under an AmScope 40X-2500X LED Lab Compound microscope, and photographed using the AmScope 5.0 MP USB Still Photo & Live Video Microscope Imager Digital Camera 5MP, manufactured by Iscope Corp., Irvine, CA, USA.

#### Immunohistochemistry for Ki-67

The Ki-67 is a chromosome-associated protein present during division (G_1_, S, G_2_, and M phases but absent from cells at rest, G_0_). Sections from paraffin-embedded hippocampal blocks were incubated for epitope retrieval in citrate buffer, pH 6.0, at 90 °C for 40 min, followed by incubation in an endogenous peroxidase blocking reagent, 0.6% H_2_O_2_ in Tris-buffered saline (TBS)-Triton (0.05% Triton X-100 in TBS, pH 7.4) for 30 min at room temperature. Thereafter, sections were pre-incubated in 2% serum (normal goat serum) + 0.1% bovine serum albumin (BSA) + 0.25% Triton in TBS for 60 min at room temperature. Afterwards, sections were incubated with polyclonal rabbit-anti-lyophilized-Ki-67p antibody (Novocastra, Newcastle, UK; 1:5000 in preincubation solution) overnight at 4 °C. Incubation with biotinylated goat anti-rabbit IgG (1:1000 + 2% normal goat serum + 0.1% BSA in TBS; Vector Lab, CA, USA; 1:250) was performed for 2 h at room temperature followed by incubation with streptavidin–biotin complex (Vectastain Elite ABC kit) and stained with 3,3′-diaminobenzidine (DAB) as chromogen. Until incubation with primary antibody, all rinses in between incubations were made with TBS-Triton, and afterwards with TBS alone.

### 2.9. Statistical Analysis

Data from the morphometry, behavior and biochemical assays were analyzed using one-way analysis of variance (ANOVA) and subjected to post hoc Bonferroni’s multiple comparison test. The results are expressed as mean ± SEM. Statistical analyses were performed using Graphpad Prism software (version 5.0, La Jolla, CA, USA). Values of *p* ≤0.05 were considered statistically significant.

## 3. Results

The exposures to both DDVP and CPF in the present study resulted in differential effects on indirect metabolic markers (body weight, brain weight and brain-body weight ratio), AChE activities, ROS levels, NO levels, histoarchitecture an distributions of proliferative nuclei proteins in the hippocampus and the amygdala, and the anxiety-related behaviors, fear learning, and spatial working memory in the exposed rats.

### 3.1. Morphometric Changes Following Exposure to DDVP and CPF

Subchronic exposures to 1/10 ratios of the oral highest tolerable dosages of both CPF and DDVP, separately and in combination, markedly caused loss of body weight over a period of 14 consecutive exposures (Figure 1A). However, the observed body weight loss was greater in the DDVP-only-exposed rats, with what may be a conflicting effect with less weight loss in the combined exposed rats (Figure 1A). There was also a significant (*p* ≤0.05) loss in brain weight of the exposed rats, with relatively more loss observed in the DDVP-only-exposed rats’ brains (Figure 1B).

### 3.2. Effects of DDVP and CPF Exposures on Spatial Working Memory

Exposures to DDVP and/or CPF significantly (*p* ≤0.05) delayed the latency to the submerged platform (escape latency) in the exposed rats in both tests for LTM (Figure 2A), STM (Figure 2B) in the MWM paradigm. Although this effect is relative to the three exposure modalities in the LTM, the combined exposures to DDVP and CPF caused a greater (*p* ≤0.05) delay in the latency to the hidden platform, followed by the DDVP-only exposure, when compared with the control (Figure 2A,B). The separate exposures to DDVP or CPF consequently resulted in avoidance (*p* ≤0.05) of the platform quadrant during the probe test for reference memory (R), while their combination surprisingly had no effect on RF (Figure 2C).

### 3.3. DDVP and CPF Exposures Increased Anxiety-Like Behavious

The latency to the closed arm, an indirect measure of fear learning, in the EPM paradigm, was significantly (*p* ≤ 0.05) delayed by exposures to both DDVP and CPF, seperately and in combination (Figure 3A) in the exposed rats. Both DDVP and CPF also caused a marked increase in freezing periods, an indication of fear, in the exposed rats. This observation was corroborated by the significant (*p* ≤0.05) reduction in time spent at the centre squares by the rats, indicating anxiety-related responses (Figure 3B,C).

### 3.4. DDVP and CPF Exposures Inhibit Anticholinesterase in the Amygdala and Hippocampus

Exposures to the two OPs used in this study, DDVP and CPF, either seperately or combined, resulted in a significant depletion in both amygdaloid (Figure 4A) and hippocampal (Figure 4B) AChE levels in the exposed rats when compared with the control’s. Although the inhibition of AChE activities in both brain regions are in relative patterns, the basal (control) AChE activity was greater in the hippocampal region. Thus, the inhibiting effects of the OPs on the hippocampal may be greater.

### 3.5. Effects of DDVP and CPF Exposures on Oxidative Stress Markers (ROS and NO) in the Amygdala and Hippocampus

Consecutive oral DDVP and/or CPF exposure in rats caused a relative (*p* ≤ 0.05) increase in both nitric oxide (NO) and total reactive oxygen species (ROS) levels in the amygdala and the hippocampus of the exposed rats (Figure 5A–D). Although no marked differences were observed in the pattern of the effects on both NO and ROS levels, CPF exposure did not result in a significant change in the hippocampal ROS level (Figure 5D).

### 3.6. Effects of DDVP and CPF Exposures on the Distributions of Proliferative Nuclei (Ki67) in the Hippocampus and the Histoarchitecture

Histological Nissl granulation stain revealed no marked effects on either the connus ammonis regions (CA1 and 3) and the dentate gyrus following exposures to DDVP, CPF, or combined exposure when compared with the control (NS). However, there were qualitatively more glia-like small-sized intensely stained cells in the DDVP- and/or CPF-exposed CA regions and the dentate gyrus (glia activation) (Figure 6), with some vacuolations mostly in the dentate gyrus (DG) of the exposed rats. Furthermore, there was a reduced presence of immunoreactive nuclei proteins of the proliferative cell marker Ki67 in CA1 and 3, as well as the DG of the DDVP- and/or CPF-exposed rats, especially in the subgranular zone of the dentate gyrus (Figure 7).

## 4. Discussion

Organophosphates poisoning account for a high percentage of reported toxicities from chemical exposure around the world, posing growing threats to public health, and with more concerns as they are continuously deposited in water bodies and the biomes [36,37]. Toxicities from these substances are primarily linked to the irreversible inhibition effects on acetylcholinesterase (AChE) in the blood and the nervous systems, thus having the ability to affect general body functions and personality-related functions [8,9,36,37,38]. In the present study, sub-chronic oral exposures to the two most commonly used broad spectrum OPs worldwide (individually and in combination) were sufficient to markedly deplete the levels of AChE in the hippocampus and the amygdala, in a pattern relative to what we recently found with CPF-only exposure on the amygdala, and with the dichlorvos in discrete brain regions, including the cerebellum, hippocampus, frontal cortex, medulla, spinal cord, and occipital cortex [9,38]. This is no surprise, as it further confirms the earlier established mechanism of OP activities in the brain. In separate studies in the literature, DDVP and CPF have been reported to cause significant inhibition of AChE in the brains of rats [8,9,39,40], of which most of its induced toxicities have been attributed.

However, there is growing evidence suggesting that although AChE inhibition contributes greatly to the toxicities and remains the primary mechanism of action of OPs, its effects on redox processes, antioxidant functions, and lipid peroxidation are greatly implicated in the chronic outcomes following poisoning [8,9,10,11,12,38]. Exposing rats to one-tenth of the LD50 oral dosages of DDVP, CPF, and their combination in the present study caused a significant increase in total reactive oxygen species (ROS) and nitric oxides (NO) levels in the hippocampus and amygdala of the exposed rats. Further corroborating previous findings on the activities of OPs on anti-oxidant defense and on general oxidative functions, and more than the AChE inhibition, these are very much implicated in the neurotoxic effects of OPs poisoning, including the neuro-cognitive impairments and cell death [9,38,41,42,43]. The oxidative damage following exposure to OPs may further contribute to its detrimental effects on health, as it has been linked to loss of biological functions in cells, contributing to the pathophysiological factors for various life-threatening diseases, like respiratory, cardiovascular, and renal diseases, carcinogenesis, and neurodegenerative disorders [3,4].

It is expected that the induced AChE dys-homeostasis and most importantly, the oxidative dysfunctions, may affect metabolic functions, since it has been implicated in different metabolic related diseases [3,4,11]. Thus, we recorded the changes in body weight at the initiation and termination of the experiment, and this revealed a significant loss in body weight, supported by a subsequent low brain weight following exposures to DDVP, CFP, and their combination, with more effects observed with DDVP exposure. These findings are affirmed by previous findings, where a loss in both body and brain weights were recorded following exposures to different OPs, including CPF and DDVP [9,38,44,45,46]. It was however rather unexpected that a combination of the two chemicals would not produce significantly greater adverse effects than uptake of the chemicals individually. This could be related to the fact that their mechanisms of action were rather similar since they targeted the same enzymes to produce the similar effects. It may have had to do with chemical interaction between the two OPs, but we did not pursue this and we believe that it remains available for further research.

An observation into the possible effects of these substances on neural functions and survival related proteins, and structures, revealed a consequent qualitative depletion of the proliferative nuclei marker (Ki67 proteins) in the hippocampal CA regions and the dentate gyrus of the DDVP, CPF and combined exposed rats. This was complemented by the observed increase in intensely stained nuclei-like glia, most especially in the dentate gyrus. These suggest possible damaging effects on potential neurogenesis and a buildup of a possible shut down of regenerative activities in the brains of the exposed rats. This can be strengthened with findings from previous studies, where exposures to neurotoxic compounds have reported to result in mark loss neurogenic cells in laboratory rodents [47,48]. Our previous examination of the effects of CPF exposure on amygdala AChE activities, oxidative markers, and expression of Ki67 proteins further supports these findings [9].

A healthy hippocampus with preserved neurogenesis is linked to enhanced psycho-cognitive functions, while any damage that affects this has been claimed to affect cognitive activities [49]. Thus, we investigated possible effects on anxiety-like behaviors and spatial working memory in the exposed rats. In congruence with the above, sub chronic exposures to either of DDVP and/or CPF significantly increased anxiety-like behaviors and impaired spatial working memory behaviors respectively. These dysfunctions in psychosocial-related and cognitive functions following exposure to the two OPs used in this study cannot be unrelated to the combined effects of the observed oxidative damage, weight loss, and diminished proliferative nuclei in the hippocampus and the amygdala. These can be strongly supported by the relative neuro-cognitive deficits that follow exposures to different insecticidal compounds including OPs [9,17,43,50,51].

OP use in human activities is likely to persist owing to evident success of such use in improving agricultural yields as well as the prevention and control of certain infectious disease vectors such as mosquitos and helminths [26]. Guidelines for the handling and use of these chemicals are widespread, however gross negligence and deliberate breaking of such rules leads to accidental exposure to the skin, eyes or even in food substances that have not been properly cleaned. Such chronic low dosages that are frequently asymptomatic in the acute phase, as those used in this study are now known to be particularly damaging [26,52]. Epidemiological studies in humans have linked OP exposure to an increased risk of developing Alzheimer’s disease [25]. Other studies have been carried out to try and establish a link between OP exposure and the incidence of neurodegenerative diseases such as Alzheimer’s disease (AD) and Parkinson’s disease (PD), and have established a collection of symptoms collectively known as COPIND [13,14,15]. Some of these results however could not establish a cause–effect relationship and are therefore inconclusive and still subject to debate and further research [25,26,52].

It is also important to note that the dosages of the DDVP and CPF used in this study were sufficient to cause oxidative and cholinesterase dys-homeostasis in the hippocampus and amygdala of the exposed rats, which subsequently resulted in marked impairments to learning and memory functions in the rats. These dosages may have been a bit high and therefore did not adequately represent normal non industrial and non-occupational exposures, but this resulted in no mortality. We noted that other authors have used similar or even higher dosages, and for longer exposure periods in some cases [53,54,55,56]. We believe that higher dosages and longer exposures are probably more representative of the human condition in the case of occupational exposure on farms and in the industrial setting.

Furthermore, to strengthen the neurocognitive findings from this study, OPs, including DDVP and CPF, have been extensively used to induce neuroinflammation (through overproduction of NO), oxidative damage (NO and ROS), polymorphisms, epigenetic dysregulation, and genetic mutations, events that are all implicated in OP-induced neurotoxicity and as underlying mechanisms for neurodegenerative diseases, including Alzheimer’s and Parkinson’s disease, among others [30,52,57,58,59,60,61,62].

## 5. Conclusions

In conclusion, sub chronic oral exposures to DDVP and CPF, separately or in combination, imposed hippocampal and amygdala oxidative damage and subsequent depletion of neurogenic nuclei in the hippocampus and dentate gyrus. These might have contributed to the psycho-cognitive deficits and increased anxiety-like behaviors that were observed following AChE inhibitions in the studied brain regions.

## Figures and Tables

**Figure 1 toxics-06-00071-f001:**
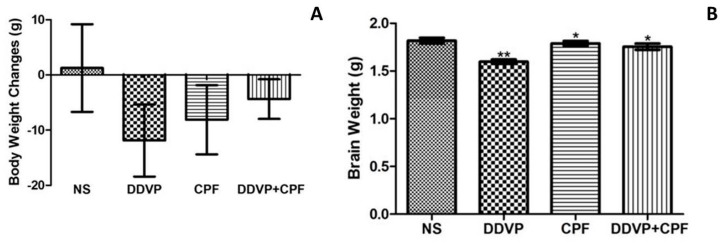
Exposures to dichlorvos (DDVP) and chlorpyrifos (CPF) results in loss of body and brain weight (**A**) Body weight of control and exposed/treated rats (**B**) Brain weight of control and exposed/treated rats. Double asterisks (**) indicate a significant (*p* ≤0.05) reduction when compared with all groups, while a single asterisk (*) indicates a significant (*p* ≤ 0.05) when compared with normal saline (NS). One-way analysis of variance (ANOVA) with a post hoc Bonferroni’s multiple comparison test.

**Figure 2 toxics-06-00071-f002:**
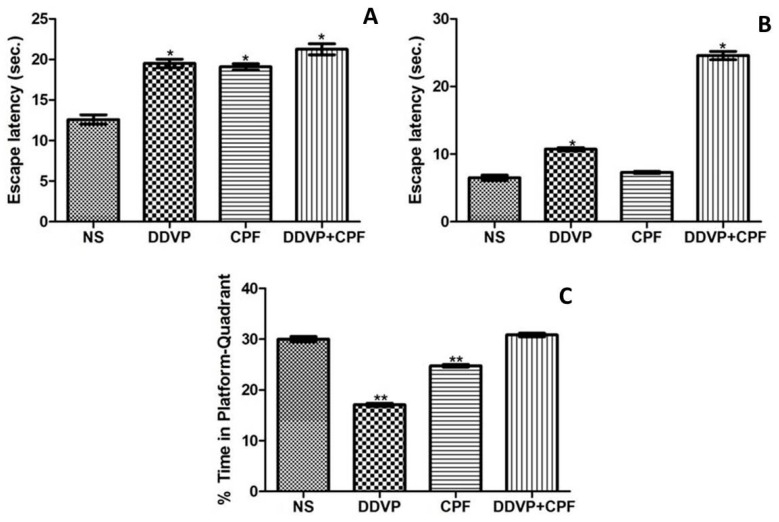
Exposure to DDVP and CPF impaired long-term memory (LTM), short-term memory (STM), and reference memory (**A**) Long-term memory (escape latency); (**B**) Short-term memory (escape latency); and (**C**) Reference memory (% time in the platform-quadrant) tests in the Morris water maze (MWM) paradigm. Double asterisks (**) indicate a significant (*p* ≤0.05) reduction when compared with NS and DDVP+CPF rats (Figure 2C), while a single asterisk (*) indicates a significant (*p* ≤0.05) increase when compared with NS (Figure 2A) and or CPF (Figure 2B).

**Figure 3 toxics-06-00071-f003:**
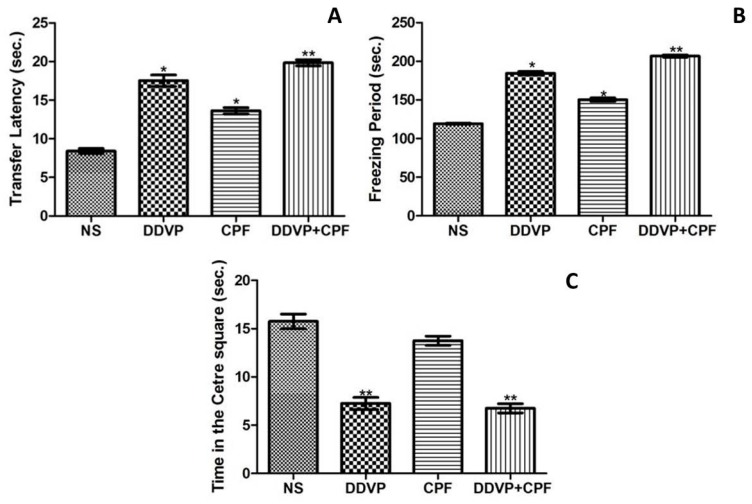
The effects of oral exposures to normal saline (NS), dichlorvos (DDVP) or/and chlorpyrifos (CPF) on: (**A**) fear learning (transfer latency) in the elevated plus maze paradigm; and (**B**,**C**) anxiety-related behavious (freezing period and time in center squares) in the open field test paradigm. Double asterisks (**) indicate a significant (*p* ≤0.05) increase (**A**,**B**) or decrease (**C**) when compared with NS, other groups, and/or CPF rats only, while a single asterisk (*) indicates significant (*p* ≤0.05) increase when compared with NS (**A**,**B**). One-way analysis of variance (ANOVA) with a post hoc Bonferroni’s multiple comparison test.

**Figure 4 toxics-06-00071-f004:**
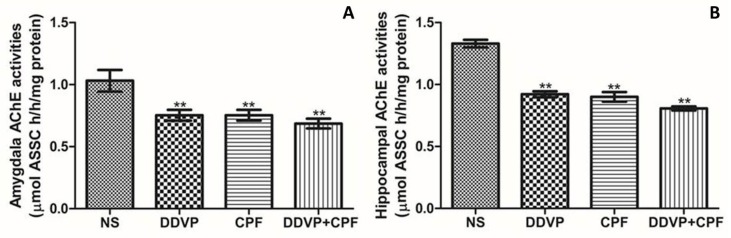
The effects of oral exposures to normal saline (NS), dichlorvos (DDVP), or/and chlorpyrifos (CPF) on: (**A**) amydaloid acetylcholinesterase (AChE) activities; and (**B**) hippocampal AChE activities in the exposed rats. Double asterisks (**) indicate a significant (*p* ≤ 0.05) decrease when compared with the NS rats. One-way analysis of variance (ANOVA) with a post hoc Bonferroni’s multiple comparison test.

**Figure 5 toxics-06-00071-f005:**
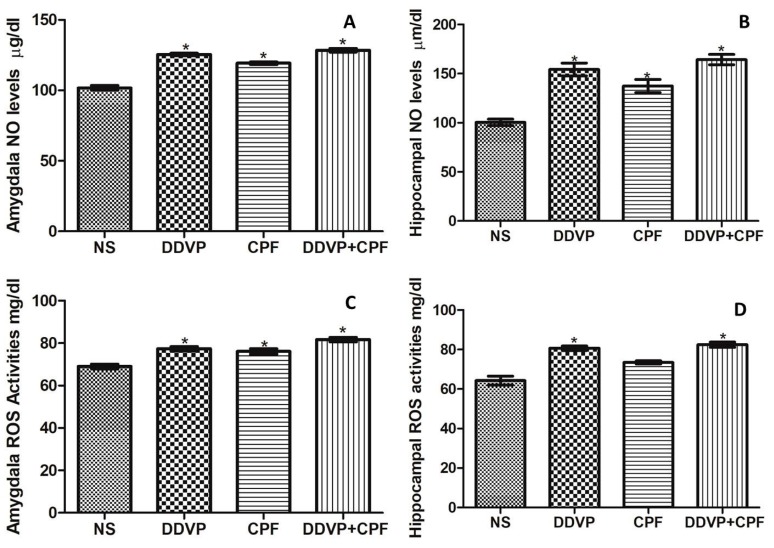
The effects of oral exposures to normal saline (NS), dichlorvos (DDVP) or/and chlorpyrifos (CPF) on: nitric oxide (NO) levels (**A**: amydala and **B**: hippocampus); and reactive oxygen specieies (ROS) levels (**C**: amygdala and **D**: hippocampus) in the exposed rats. Single asterisk (*) indicates a significant (*p* ≤0.05) increase when compared with the NS rats. One-way analysis of variance (ANOVA) with a post hoc Bonferroni’s multiple comparison test.

**Figure 6 toxics-06-00071-f006:**
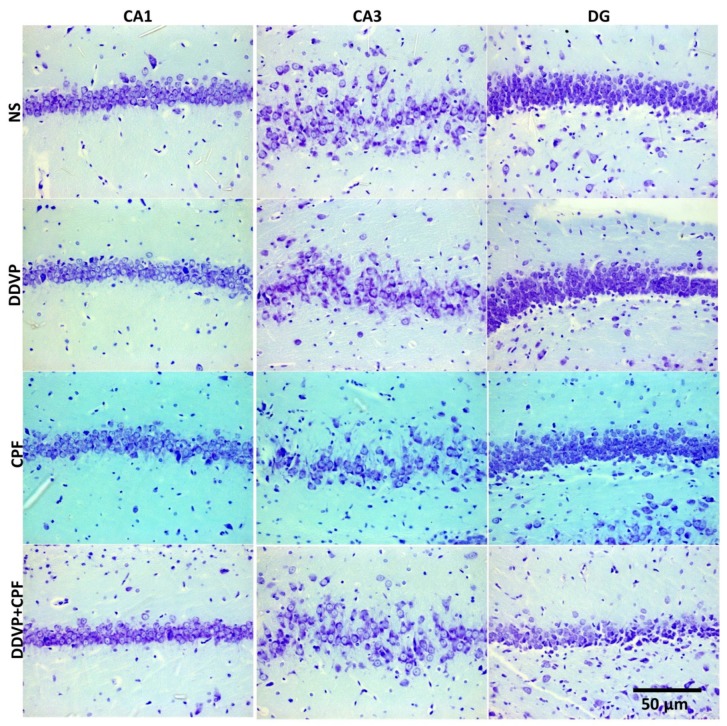
The effects of oral exposures to normal saline (NS), dichlorvos (DDVP), or/and chlorpyrifos (CPF) on: the hippocampal connus ammonis 1 and 3 (CA1 and 3), and the dentate gyrus (DG) in the exposed rats. There was no marked changes following either DDVP and/or CPF when compared with the control (NS). Scale bar 50 µm.

**Figure 7 toxics-06-00071-f007:**
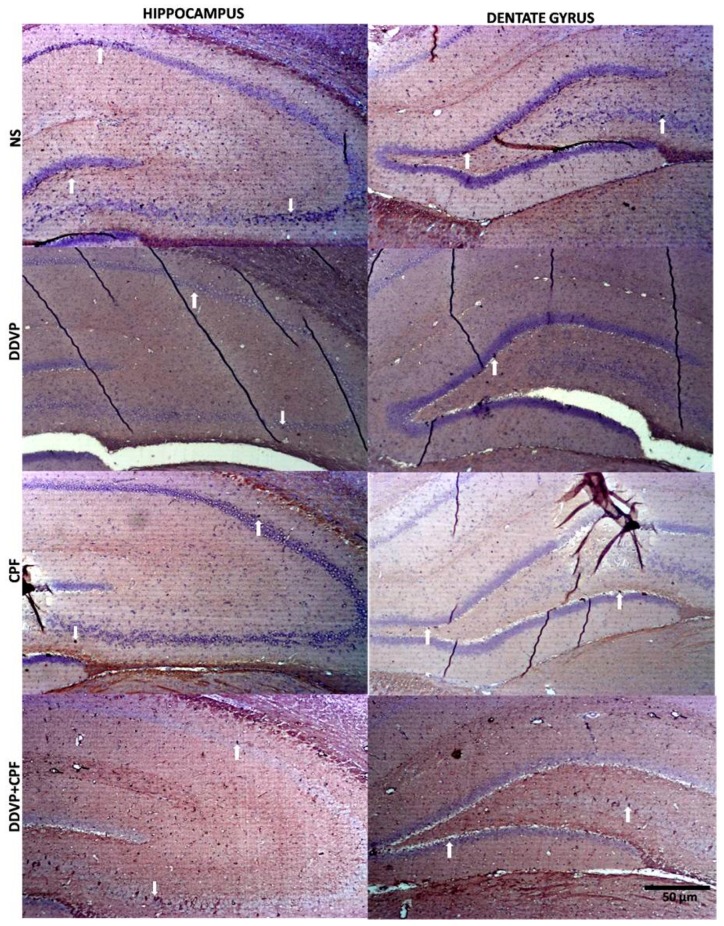
The effects of oral exposures to normal saline (NS), dichlorvos (DDVP), or/and chlorpyrifos (CPF) on: the distributions of Ki67 nuclei proteins in the hippocampal connus ammonis regions (CA1 and 3), and the dentate gyrus (DG) in the exposed rats. White arrows indicate the Ki67 immunoreactive proteins in the respective regions, with reduced nuclei in the DDVP and/or CPF exposed rats compared to the control. Scale bar 50 µm.

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
