# Peer review of "Chlorpyrifos- and Dichlorvos-Induced Oxidative and Neurogenic Damage Elicits Neuro-Cognitive Deficits and Increases Anxiety-Like Behavior in Wild-Type Rats"

_toxics, 2018, doi:10.3390/toxics6040071_

Round 1
Reviewer 1 Report
In this study, authors investigated in a rat model the neuronal impact of an oral exposure to two organophosphorus pesticides Chlorpyrifos (CPF) and dichlorvos (DDVP). Male rats were exposed during 14 days by gavage to CPF and DDVP at 14.9 and 8.8 mg/kg body weight respectively. Various parameters linked to neuro-cognitive performance and anxiety like behaviors were assessed upon a 10 day-period of treatment. At the end of the experimental period the rats were euthanized, and nitric oxide (NO) metabolites, total reactive oxygen species (ROS), and acetylcholinesterase (AChE) activity, as well as histology and immunohistochemistry were performed in hippocampus and amygdala . Authors aimed in this study, to provide an insight on the role of oxidative and neurogenic damages on the neurogenic dysfunction and increased anxiety-like behavior induced upon organophosphate poisoning.
Although this study matched to the growing societal demand of improving knowledge on the health impact of pesticide , this article has imperfection and the originality of the results and conclusions compared to the various previous published data on the same topic does not appears clearly . Moreover, conclusions on the link between biochemical events in some cerebral regions and psycho-cognitive defects or increased anxiety-like behaviors in response to pesticide exposure are not well supported by the results. Thus, this article cannot be published in this present form and major points need to be revised:
1) Introduction : originality of the project does not appear clearly: (i)many studies have already shown the neurotoxic effect of organophosphate compounds (OP) (ii) many studies have reported the impact of these compounds on oxidative status (iii) thus the question is : what is the originality of the present study compared to the numerous previous published data? Efforts are needed in the introduction to provide sufficient background
2) Introduction : authors must better explain why they have focused their interest on the hippocampus and amygdala brain regions ;
3) Introduction: As poisoning upon OP exposure is evoked in the study, authors must compared the dose assessed in their study with the circulating dose of OP in human upon poisoning. Authors must also specify the relevance of the chosen dose and must give the toxicological reference value of the compounds of interest.
4) Introduction: Authors must specify why they focused only on two OP.
5) Material and method : methods are adequately described ; authors must explain why they focused their study only in male; comparison of the effect of OP between male in female would improve the study
6) Analysis of biochemical parameters in serum (ALAT, ASAT) are needed to have an idea of the degree of toxicity of the treatment
7) Figure 1 brain weight must be expressed as g per g of body weight
8) Figure 1 authors reported a markedly loss of body weight upon exposure to pesticide alone or in combination: however, no statistical difference appears in figure 1.
9) Since changes reported by authors are often slight, authors must specify whether the observed changes (brain weight, reference memory, shot term memory, transfer latency, ROS and NO production etc. and those described in figures 3, 4, 5, 7) are relevant by adding a positive control group for example.
10) As doses are very high, do the amplitude of the observed changes upon exposure to pesticides are relevant and have severe consequences?
11) As mixture effects are not always higher than effects of pesticide alone according to the assessed parameter, authors must discuss this point
12) The cause and effect relationship between oxidative and neurogenic slight changes and neuro-cognitive performance and anxiety-like behavior is not proven in this study. Authors described comportment disorders at day 10 of treatment and biochemical perturbations at day 14. This does not prove that these two phenomena are linked together.
Author Response
Chlorpyrifos- and dichlorvos-induced oxidative and neurogenic damage elicits neuro-cognitive deficits and increases anxiety-like behaviors in wild-type rats
List of corrections
Reviewer #1 | |
Comments | Responses |
Thus the question is : what is the originality of the present study compared to the numerous previous published data? Efforts are needed in the introduction to provide sufficient background
| The study focuses on the effect of combined dosage of 2 distinct OP substances. The study also specifically looks at the behavioural changes associated with the simultaneous uptake of both substances as well as changes in specific areas of the brain (hippocampus and amygdala) that could be could these behavioural alterations |
Authors must better explain why they have focused their interest on the hippocampus and amygdala brain regions | The hippocampus and amygdala are 2 brain regions that are known to be directly involved in spatial memory and emotional reactions, such as fear. The battery of behavioural tests used in this study specifically tested for spatial memory and amygdala dependent behaviours. It therefore followed that should there be any behavioural changes observed, our first port of call would be the brain regions known to affect such behaviours. |
As poisoning upon OP exposure is evoked in the study, authors must compared the dose assessed in their study with the circulating dose of OP in human upon poisoning. Authors must also specify the relevance of the chosen dose and must give the toxicological reference value of the compounds of interest. | Previous studies have shown that using a tenth of LD50 provides a fair estimate when toxicology studies in rats are projected to the human condition “Newell DR, Burtles SS, Fox BW, Jodrell DI, Connors TA. Evaluation of rodent-only toxicology for early clinical trials with novel cancer therapeutics. British journal of cancer. 1999 Nov;81(5):760.” This has inserted in the methods section. |
Authors must specify why they focused only on two OP | Previous research has shown that these are 2 of the most commonly used OPs |
Authors must explain why they focused their study only in male; comparison of the effect of OP between male in female would improve the study | Normally in studies of natural populations, ethics committees recommend the use of male species as females are key in replenishing numbers after sacrifice. We also felt that due to the influence of fluctuations in circulating levels of sex hormones that do have some recognized neuroprotective functions, OPs in the female rats could a different paper altogether |
Analysis of biochemical parameters in serum (ALAT, ASAT) are needed to have an idea of the degree of toxicity of the treatment | In a previous study from our group, the effects of 8.8mg/kg DDVP on liver toxicity markers (ALT, ALP, AST and GGT) were documented (Ajao, M. S., Abdussalam, W. A., Imam, A., Amin, A., Ibrahim, A., Adana, M. Y., ... & Atata, J. A. (2017). Histopathological and Biochemical evaluations of the antidotal efficacy of Nigella sativa oil on organophosphate induced hepato-toxicity. Research Journal of Health Sciences, 5(1), 18-25). However the present study only focused on oxidative stress and effects on behavioral change |
Figure 1 brain weight must be expressed as g per g of body weight | Effected |
Figure 1 authors reported a markedly loss of body weight upon exposure to pesticide alone or in combination: however, no statistical difference appears in figure 1. | Effected |
Since changes reported by authors are often slight, authors must specify whether the observed changes (brain weight, reference memory, shot term memory, transfer latency, ROS and NO production etc. and those described in figures 3, 4, 5, 7) are relevant by adding a positive control group for example | Thank you for this comment. The authors appreciate the fact that a positive control is important in toxicity studies. We feel that the single OP treated groups (CPF only and DDVP only) serve as positive controls for the effect |
As doses are very high, do the amplitude of the observed changes upon exposure to pesticides are relevant and have severe consequences? | The selected dosages of both chemicals were 1/10 of the LD50 obtained from previous studies, we believe increasing these dosages could increase the amplitude of observed changes |
As mixture effects are not always higher than effects of pesticide alone according to the assessed parameter, authors must discuss this point | A section discussing this point has been introduced |
The cause and effect relationship between oxidative and neurogenic slight changes and neuro-cognitive performance and anxiety-like behavior is not proven in this study. Authors described comportment disorders at day 10 of treatment and biochemical perturbations at day 14. This does not prove that these two phenomena are linked together | It probably does not prove a cause and effect relationship, but it does suggest a link or correlation between the two. The best would have been to sacrifice the animals immediately after behavioural tests to check for biochemical and histochemical changes. However, this was not possible since we had a battery of tests for the animals before sacrificing them. |

Reviewer 2 Report
I consider the manuscript of Imam and coworkers of interest for scientific community. The work shows some important neurotoxicity effects in rat brain after exposure to the OP.
However I have some suggestions and some minor concerns listed below.
The authors can follow my suggestions about few incorrect sentences and mistakes (see pdf attached with revision).
I strongly suggest improving the morphological and histological data to better define the effects of OP on different brain regions.
- In particular, authors may define better in Fig. 6 the area where glial-like small cells are present. Moreover, a statistical analysis must be performed to count the numbers of these glial-like cells in the NS, DDVP, CPF and DDVP+CPF. For this purpose, authors could use for example ImageJ.
-also for Ki67 labeling (Fig.7), the authors must add new Figures with a higher magnification to better show Ki67 immunoreactive nuclei in different treatment conditions as in control.

Author Response
Manuscript
Chlorpyrifos- and dichorfos-induced oxidative and neurogenic damage elicits neuro-cognitive deficits and increases anxiety-like behaviors in wild-type rats
List of corrections
Reviewer #2 | |
I consider the manuscript of Imam and coworkers of interest for scientific community. The work shows some important neurotoxicity effects in rat brain after exposure to the OP | Thank you for this comment |
I strongly suggest improving the morphological and histological data to better define the effects of OP on different brain regions In particular, authors may define better in Fig. 6 the area where glial-like small cells are present. Moreover, a statistical analysis must be performed to count the numbers of these glial-like cells in the NS, DDVP, CPF and DDVP+CPF. For this purpose, authors could use for example ImageJ. | It has been improved |
also for Ki67 labeling (Fig.7), the authors must add new Figures with a higher magnification to better show Ki67 immunoreactive nuclei in different treatment conditions as in control. | These sections are in the University of Ilorin, Nigeria where the stain was conducted, thus new images are not presently possible. We have now inserted arrows to indicate location of the Ki67 proteins |
Reviewer 3 Report
Thank you for the information!
First sentence of abstract is too "wordy." Consider revising.
Line 34: period after "rats."
Introduction: Are you primarily discussing occupational exposures instead of lower-level exposures from food, etc.? Be more explicit.
Line 131: mounted, not mount
Line 139-140: Was STM measured on a test where the platform had been moved? If the platform wasn't moved it seems to me like the rats would still be working off their long-term memory of where the platform had always been.
OFT procedure and EPM procedure: A figure would be much easier understood than words describing the set-up (particularly for those less familiar with animal testing). And I understand that the rats staying at the perimeter of the OFT set-up indicates anxiety, but you should explicitly say that.
Line 167: You reference a treat but I don't know where it was in the first place. Do you always use a treat? And shouldn't the rats be moving to the open arms, not the closed arms, since they don't like the open space?
Figure 1B: Move the y-axis up to around 1 or even 1.5 g so we can "zoom in" and see precision better.
Line 248: Aren't both LTM and STM tests part of the MWM paradigm? Remove comma after although
Line 252: Delete comma after quadrant
Line 253: "had" no effect
Figures 3 and 5: Consider not having y-axis all the way down at zero so we can see precision better.
Line 292: remove comma after rats
Line 294: Remove comma after although
Line 335: Add comma after "study"
Line 336: ...broad spectrum OPs worldwide both seperately and in combination was sufficient...
Line 347: What do you mean by oral tolerable dosage?
Line 358: remove comma after "expected"
End of discussion: Can you give us a paragraph or two extending your findings to potential effects on human populations? Of course, don't give speculations that can't be supported, but explain the significance of your work.
Author Response
Manuscript
Chlorpyrifos- and dichlorvos-induced oxidative and neurogenic damage elicits neuro-cognitive deficits and increases anxiety-like behaviors in wild-type rats
List of corrections
Reviewer #3 | |
First sentence of abstract is too "wordy." Consider revising.
| The first sentence has been revised and shortened |
Line 34: period after "rats."
| Period inserted |
Introduction: Are you primarily discussing occupational exposures instead of lower-level exposures from food, etc.? Be more explicit. | Occupational exposure has been associated more with cases of accidental poisoning than the very low level exposures. However, in our study we suggest that low level exposure that may initially appear asymptomatic may actually have dire long term consequences as well |
Line 131: mounted, not mount | Corrected as suggested |
Line 139-140: Was STM measured on a test where the platform had been moved? If the platform wasn't moved it seems to me like the rats would still be working off their long-term memory of where the platform had always been | The method that we used is a standard that has been used by others before. In previous studies, animals with no indication of oxidative stress have reached the platform faster as an indicator of better short term memory. We agree that behavioural tests are rather difficult to correctly and adequately interpret
|
OFT procedure and EPM procedure: A figure would be much easier understood than words describing the set-up (particularly for those less familiar with animal testing). And I understand that the rats staying at the perimeter of the OFT set-up indicates anxiety, but you should explicitly say that | Rather than repeat that which has already been extensively covered in the literature, we have included references that include detailed descriptions of the methods.
This has now been addressed by altering the sentence to read thus “It has been reported that avoidance of central squares with preference towards the perimeter areas provides an indication of elevated anxiety levels in the rats” |
Line 167: You reference a treat but I don't know where it was in the first place. Do you always use a treat? And shouldn't the rats be moving to the open arms, not the closed arms, since they don't like the open space? | Sentence altered to read “The time it took the rats to retreat and move to the closed arms was recorded as the transfer latency….” |
Figure 1B: Move the y-axis up to around 1 or even 1.5 g so we can "zoom in" and see precision better. | Advice accepted |
Line 248: Aren't both LTM and STM tests part of the MWM paradigm? Remove comma after although
| They both are part of the MWM, thank you for this observation, the sentence has been altered to reflect this The comma has been removed |
Line 252: Delete comma after quadrant
| The comma has been deleted |
Line 253: "had" no effect
| Change effected as suggested |
Figures 3 and 5: Consider not having y-axis all the way down at zero so we can see precision better. | That is been effected |
Line 292: remove comma after rats | The comma has been deleted |
Line 294: Remove comma after although
| The comma has been removed |
Line 335: Add comma after "study"
| Comma has been added |
Line 336: ...broad spectrum OPs worldwide both seperately and in combination was sufficient... | Edited to read “broad spectrum OPs worldwide, individually and in combination was sufficient to” |
Line 347: What do you mean by oral tolerable dosage?
| Changed to read “LD50 oral dosages” which refers to 1/10 of the dose that kills 50% of rats receiving treatment or exposure |
Line 358: remove comma after "expected"
| The comma has been removed |
End of discussion: Can you give us a paragraph or two extending your findings to potential effects on human populations? Of course, don't give speculations that can't be supported, but explain the significance of your work.
| A paragraph to this effect has been added |
Round 2
Reviewer 1 Report
Authors answered correctly to the various points. However he question remains on the relevance of the observed effects in their study since these effects appear slight ?.
Moreover, authors did not respond adequately to the following questions : " As doses used in their study are very high, do the range of the observed changes upon exposure to pesticides are relevant and have severe consequences?". Authors could answer by showing for exemple the extent of the rating changing in the case of Alzheimer disease or other pathologies in which the parameters of interest are dysregulated
Author Response
Comments and Suggestions for Authors
Authors answered correctly to the various points. However he question remains on the relevance of the observed effects in their study since these effects appear slight?.
Response
We respect the views of the reviewer, but we will like to bring the attention to the changes which are in the results following the OPs exposures, of which we derived our conclusions from:
1. About 5 to 12 grams loss in body weight
2. About 8 to 12 and 5 to 18 seconds delay in escape latency in LTM and STM tests respectively
3. Low quadrant visit for reference memory
4. About 11 to 15 seconds delay in the transfer latency in fear related memory test
5. About 60 to 90 seconds freezing behavior in the OFT, an indication of fear
6. About 30 to 40% depletion in AChE activities, and 20 to 40% increase in ROS and NO levels
7. Marked loss in neurogenic (Ki67) protein
All of the above effects are statistically significant.
Moreover, authors did not respond adequately to the following questions : " As doses used in their study are very high, do the range of the observed changes upon exposure to pesticides are relevant and have severe consequences?". Authors could answer by showing for exemple the extent of the rating changing in the case of Alzheimer disease or other pathologies in which the parameters of interest are dysregulated
Response
Thank you, we have now included a section in the concluding part of the discussion that addressed this point.
“It is also important to note that the dosages of the DDVP and CPF used in this study were sufficient to cause oxidative and cholinesterase dys-homeostasis in the hippocampus and amygdala of the exposed rats, which subsequently resulted in marked impairments to learning and memory functions in the rats. These dosages may have been a bit high and therefore did not adequately represent normal non industrial and non-occupational exposures but this resulted in no mortality. We noted that other authors have used similar or even higher dosages, and for longer exposure periods in some cases [53-56]. We believe that higher dosages and longer exposures are probably more representative of the human condition in the case of occupational exposure on farms and in the industrial setting.
Furthermore, to strengthen the neurocognitive findings from this study, OPs, including DDVP and CPF have been extensively used to induce neuroinflammation (like over production of NO), oxidative damage (NO and ROS), polymorphisms, epigenetic dysregulation and genetic mutations, events that are all implicated in OPs induced neurotoxicity and as underlying mechanisms to neurodegenerative diseases, including Alzheimer’s and Parkinson’s diseases among others [57-64].”